# Cigarette Affordability and Cigarette Consumption among Adult and Elderly Chinese Smokers: Evidence from A Longitudinal Study

**DOI:** 10.3390/ijerph16234832

**Published:** 2019-12-01

**Authors:** Xiao Hu, Yang Wang, Jidong Huang, Rong Zheng

**Affiliations:** 1School of International Trade and Economics, University of International Business and Economics, Beijing 100029, China; 201500130112@uibe.edu.cn (X.H.); 201500130006@uibe.edu.cn (Y.W.); 2School of Public Health, Georgia State University, Atlanta, GA 30303, USA; jhuang17@gsu.edu

**Keywords:** cigarette affordability, cigarette consumption, elasticity, China, GEE model

## Abstract

China is in the midst of an epidemic of non-communicable diseases (NCDs), which has increasingly accounted for a growing share of disease burden, due in part to China’s ongoing rapid socioeconomic changes and population aging. Smoking, the second leading health risk factors associated with NCDs in China, disproportionately affects the old population more than their younger counterparts. Using survey data from the China Health and Retirement Longitudinal Study (CHARLS), this study evaluated the impact of changes in cigarette affordability on smoking behavior among middle-aged and elderly (age 45 and older) smokers. Self-reported cigarette price and disposable income were used to calculate cigarette affordability. Cigarette consumption was measured using the number of cigarettes smoked per day reported by the survey respondents. The correlation between cigarette affordability and cigarette consumption was estimated using generalized estimating equations adjusting for demographics, socioeconomic status, geolocations, and cigarette price tiers, as well as year fixed effects. The estimated overall conditional cigarette affordability elasticity of demand was –0.165, implying a 10% decrease in cigarette affordability would result in a reduction in cigarette consumption by 1.65%. The cigarette affordability responsiveness differs by demographics, socioeconomic status, geolocations, and cigarette price tiers. This study provides evidence that tax/price policies that reduce cigarette affordability could lead to a decrease in cigarette consumption among middle-aged and elderly smokers in China. Smoke-free laws, as well as minimum price regulations, may be needed to compliment excise tax policy to target specific smoking subgroups whose cigarette consumption is less sensitive to changes in cigarette affordability.

## 1. Introduction

The Chinese population is ageing dramatically, average life expectancy at birth has risen from 44.6 years in 1950 to 75.3 years in 2015, and is expected to reach 80 years by 2050. Importantly, the pace of population ageing is much faster in China than many other countries, both high-income and low/middle-income ones. In the next 30 years, the percentage of people in China aged 60 years or older is expected to more than double, growing from 15.0% (211 million people) in 2015 to 31.9% (422 million) in 2050 [1,2]. The rapid aging population in China exerts a heavy burden on the country’s workforce and poses substantial challenges for China’s healthcare systems. The total expenditures for elderly care in China as a share of its gross domestic product (GDP) are projected to increase from 7% in 2015 to more than 26% by 2050 [3]. One of the potential ways to address this crisis caused by aging-related non-communicable diseases and the associated economic burden is to reduce unhealthy behaviors. Tobacco use is the second leading risk factor of preventable deaths and disabilities in China [4]. Evidence accumulated to date demonstrates that smoking reduction and cessation will decrease the smoking attributed deaths and diseases [5,6,7,8,9,10]. Although the overall smoking rate declined from 27.7% in 2015 to 26.6% in 2018 in China, the number of cigarettes consumed per day among Chinese smokers did not decrease between 2015 and 2018 [11]. In 2018, 26.6% of the Chinese adult population (age 15 and above) were smokers, and the smoking prevalence of male smokers (50.5%) was substantially higher than that of female smokers (2.1%). The prevalence is also substantially higher among middle-aged and elderly (age 45 and older) people—30.2% among people aged 45–64 years and 23.1% among people aged 65 years and older respectively. Given the well-documented relationship between smoking and the increase in mortality and morbidity caused by non-communicable diseases (NCDs), it is important for China to recognize the growing epidemic of NCDs associated with its aging society, and adopt appropriate policies to address this epidemic.

Raising cigarette prices through increasing cigarette tax is widely recognized as one of the most effective tobacco control measures [12]. However, the reduction in cigarette consumption resulting from an increase in cigarette price occurs only when the price increase reduces cigarette affordability [13]. Previous studies examined the relationship between average cigarette affordability and aggregate cigarette consumption by estimating the affordability elasticities of demand, using aggregated national level data [14,15,16]. Blecher and van Walbeek (2004), for example, investigated the relation between cigarette affordability and consumption in seventy countries between 1990 and 2001 [14]. Extending on the work of Blecher and van Walbeek, He et al. (2018) examined the association between cigarette affordability and consumption, and calculated the affordability elasticity of demand in 78 countries from 2001 to 2014 [15]. Zheng et al (2017) investigated the cigarette affordability trend and elasticities in China during the period of 2001 to 2016 by using aggregate data [16]. To our knowledge, no published studies have estimated the affordability elasticity of cigarette demand using individual micro-level data in China. While aggregate data are useful in understanding the overall impact of cigarette affordability on aggregated cigarette consumption, individual micro-level household survey data can examine whether the effect of cigarette affordability on individual cigarette consumption differ by population subgroups, defined by demographics, socioeconomic status, geographic locations and cigarette price tiers.

This paper aims to fill this critical gap by estimating the affordability elasticity of demand for cigarettes using individual level data obtained from a national representative survey in China. Specifically, we examined the associations between cigarette affordability and cigarette consumption and the potential differential responsiveness of cigarette consumption to changes in cigarette affordability by employing the individual-level survey data—the China Health and Retirement Longitudinal Study (CHARLS)—in China. It focuses on how daily cigarette consumption responds to changes in cigarette affordability among middle-aged and elderly Chinese smokers, a group known to be more likely to be affected by NCDs compared to their younger counterparts [17]. A better understanding of the relationship between cigarette affordability and daily cigarette consumption, particularly among middle-aged and elderly smokers, can help inform the discussion and the adoption of appropriate policies aiming to reduce cigarette smoking, and consequently the cost of NCDs in China.

## 2. Materials and Methods 

### 2.1. Data Source

The data used in this paper came from the China Health and Retirement Longitudinal Study (CHARLS). CHARLS is a nationally representative longitudinal cohort study of adults aged 45 years and older living in mainland China (including 28 provinces, municipal cities and autonomous regions). A multi-stage stratified sampling design was used to ensure the representativeness of the sample. The detailed description of the sampling method, quality assurance measures and the questionnaire were published elsewhere [18]. Using a face-to-face computer-assisted person interview (CAPI) format, the baseline (Wave One) of CHARLS surveyed 17,708 participants between June 2011 and March 2012; 18,605 participants (including 2834 replenish participants) participated the first follow-up (Wave Two) survey in 2013 and 21,095 participants (including 574 replenish participants) were surveyed at the second follow-up (Wave Three) survey between July 2015 and January 2016.

Wave One and Three of CHARLS were used in this study. Wave Two was not used in this study due to missing data on key outcome and explanatory variables among a large proportion of smokers, as well as the inconsistency in measuring per capita disposable income as explained below.

### 2.2. Measures

#### 2.2.1. Per Capita Disposable Income

Annual per capita household income was used to measure per capita disposable income in our study. Total household income was captured by wage income of each household member, outside-originated private money transfer for each household member, household agricultural income, net income of self-employed activities, household public transfer income, monetary support from non-coresident family members and other household income. Per capita disposable income was then constructed as the total household income divided by the number of household members.

The household members included: i) people who live together and share daily expenses (including parents supported by this household); ii) those who do not currently live in this household, but live in dormitories, workplaces or other impermanent places and iii) live-in nannies, drivers or any other service people. Meanwhile, those related individuals who live separately with the main respondent for more than 1 month in the past year were excluded. However, those who have been away from home for more than 1 month were treated as a household member in Wave Two, and there was no way to eliminate them. For this reason, as well as the significant missing data issue, Wave Two data were excluded from this analysis.

#### 2.2.2. Cigarette Affordability and Consumption

The key measure in this study was cigarette affordability, defined as the relative income price (RIP) of factory-made cigarettes (include filtered cigarettes and unfiltered cigarettes). In all waves of CHARLS, smokers reported the cigarette price per pack (20 sticks) that they paid for the cigarettes they consumed. The RIP was constructed using two different measures. The first RIP measure was constructed as the percentage of per capita household disposable income required to purchase 100 packs of 20-stick cigarettes (measure #1 of RIP). Higher values of RIP indicate lower cigarette affordability, and vice versa. The rationale for using this measure was based on the consumer behavior theory, which outlines various factors, including psychological, socio-economic, demographic and personality characteristics, that could influence the personal purchase intentions [19]. Consequently, the household characteristics, such as family size, and burden of raising children and supporting parents, play an important role in personal willingness to consume, especially in China [20]. As a result, per capita household disposable income could better explain consumers’ choice than personal income. However, the per capita household disposable income may not capture accurately the disposable income available to the smokers in a household since the income was assumed to be equally shared among those living in the same household. As such, we developed a second RIP measure (measure #2 of RIP) by using personal income instead of per capita household disposable income to test the robustness of our baseline results. Personal income was captured by wage income (for those who still works) and pension income (for those retired). For those unemployed, self-employed and farming workers, however, there was no personal wage or pension income available and we were unable to identify their contribution to the total household income. As a result, the analytical sample size that uses the measure #2 was reduced.

In this study, we used the number of cigarettes consumed per day (CPD), reported by smokers, to measure the daily cigarette consumption, or smoking intensity. 

#### 2.2.3. Covariates 

We included demographic, socioeconomic characteristics and geographic locations of respondents in the analysis. Demographic characteristics considered in this study included gender (male and female), age category (45–59, 60 and above) and marital status (married and unmarried). Socioeconomic covariates included educational level (primary school and below, middle and high school and college and above). Geolocations included urban-rural indicator and region indicator [21] (the Northeast region: Heilongjiang, Jilin and Liaoning; the Central region: Shanxi, Henan, Hubei, Hunan, Jiangxi and Anhui; the Eastern region: Beijing, Tianjin, Hebei, Shandong, Jiangsu, Shanghai, Zhejiang, Fujian and Guangdong; the Western region: Chongqing, Sichuan, Guangxi, Guizhou, Yunnan, Shaanxi, Gansu, Inner Mongolia, Xinjiang and Qinghai).

To allow for the effect of variations in cigarette brands and quality, we further controlled for the price tiers of cigarettes purchased by participants. Cigarette brands in China are categorized into five price tiers (or classes) by the State Tobacco Monopoly Administration (STMA) based on the allocation price (or produce price): Tier I (≥ 10 RMB/pack); Tier II (7–10 RMB/pack); Tier III (3–7 RMB/pack); Tier IV (1.65–3 RMB/pack) and Tier V (< 1.65 RMB/pack). After converting the STMA’s produce price classification into retail price classification (Tier I (≥ 20 RMB/pack) luxury brands; Tier II (13–20 RMB/pack) premium brands; Tier III (6–13 RMB/pack) medium-priced brands; Tier IV (3–6 RMB/pack) discount brands, and Tier V (< 3 RMB/pack) deep discount brands.), we constructed the price tier variable to identify the cigarette brand category consumed by participants based on the cigarette price they reported.

### 2.3. Study Sample

For both Wave One and Wave Three data, we excluded the top and bottom 2.5 percentile of responses on the disposable income variable to minimize the distortions caused by the outliers [22]. Responses on cigarette price with extremely low values (less than 1 RMB/pack) were also excluded from the analysis. In addition, respondents who reported spending more than 100% of their income on cigarettes, and those with missing data on the covariates were also excluded from the analysis. The final analytical sample consisted of 6652 observations on 5077 individual smokers, with 1575 individuals existing in both Wave One and Wave Three. Besides, the summary statistics of key indicators, including RIP, CPD, cigarette price and per capita household disposable income, are presented in Table A1 and Table A2.

### 2.4. Empirical Methodology

Relying on the pooled dataset from Wave One and Wave Three of the CHARLS, we used Equation (1) and Equation (2) to analyze the patterns in cigarette affordability and daily cigarette consumption, respectively.
(1)$$RIP_{it}=\alpha_{0}+\alpha_{1}X_{it}+w_{i}+e_{it},$$
(2)$$Cig\_consumption_{it}=\beta_{0}+\beta_{1}X_{it}+w_{i}+v_{it},$$
where i denotes individual and t denotes year. $RIP_{it}\;$is the cigarette affordability for the $i_{th}$ individual in survey wave/year t. $Cig\_consumption_{it}$ is the average number of CPD by the $i_{th}$ individual in survey wave/year t. $X_{it}$ is a vector of individual demographics (age, gender and marital status), socioeconomic characteristics (education level), geographic indicators (urban/rural status and economic regions) and cigarette price tiers. $w_{i}$ is survey wave/year dummy variable for $i_{th}$ individual.

Equation (3) was used to examine the correlation between per capita cigarette consumption and cigarette affordability.
(3)$$ln\left(Cig\_consumption_{it}\right)=\gamma_{0}+\gamma_{1}ln(RIP_{it})+\gamma_{2}X_{it}+w_{i}+u_{it}.$$

To test whether there were any differential affordability responsiveness by age, gender, education, marital status, urban/rural status, economic regions and cigarette price tiers, a set of interaction terms ($RIP_{it}\times x_{it}$) were added into Equation (3), with each interaction term entering Equation (3) separately and independently (see Equation (4)).
(4)$$ln(Cig\_consumption_{it})=\delta_{0}+\delta_{1}ln(RIP_{it})+\delta_{2}X_{it}+\delta_{3}ln(RIP_{it})\times x_{it}+w_{i}+\mu_{it},$$
where $x_{it}$ in Equation (4) is one of the covariates in $X_{it}$.

Since the CHARLS data are longitudinal, errors are correlated within two observations for the same individual. Hence, equations were estimated using generalized estimating Equation (GEE) method (STATA version 15 xtgee command) that fits population-averaged panel-data model. Due to the nature of the cigarette affordability and consumption variables, which are continuous, Equations 1–4 were estimated using the GEE model with an identity link. In addition, we adopted the quasilikelihood under the independence model criterion (QIC) method to select the best-working correlation structure [23]. When comparing two or more different correlation structures for a specified distribution and link function, a correlation structure with the smallest QIC is the preferred correlation structure [24]. A smaller QIC was achieved by using the independence correlation structure compared to the exchangeable correlation structure, and as a result, the former was chosen as the preferred correlation matrix.

## 3. Results

### 3.1. Sample Characteristics

Table 1 summarizes the characteristics of the study population, which was fairly evenly distributed among age groups, with 53% of the smokers aged 45–59 years and 47% aged 60 years and older. The smokers in the sample were predominantly male (91%). Fifty-five percent of smokers reported having less than primary school education, 43% reported having completed middle or high school and only 3% of smokers reported having a college degree or other professional degrees. Ninety percent of smokers were married at the time of the survey. Thirty-one percent of the smokers in the sample resided in the Central region, 32% in the Eastern region, 29% in the Western region and the Northeast region accounted for only 8% of the smokers. In addition, about 58% of smokers were rural residents and 42% were urban residents. The cigarette brands used by most of the smokers belonged to the Tier III (medium-priced brands, 30%) and Tier IV (discount brands, 48%). Only 6% of the smokers reported using luxury and premium brands (Tier I and II), and 16% of smokers reported using the deep discount brands (Tier V).

### 3.2. Cigarette Affordability and Cigarette Consumption

Table 2 presents the GEE estimates based on multivariable analysis outlined in Equation (1) and (2). The results in Table 2 show that there are significant differences in cigarette affordability and cigarette consumption per day across different subgroups of smokers, defined by demographics, socio-economic status, location of residence and cigarette price tiers. Specifically: (i) cigarette affordability was significantly lower among smokers whose age was 60 years and older (RIP 1.60 percentage points (pp) higher). In addition, elderly smokers aged 60 and above consumed fewer cigarettes per day (CPD 2.41 sticks fewer) than those smokers aged 45–59 years; (ii) while cigarette affordability was not statistically different between female and male smokers, daily cigarette consumption was significantly lower among female smokers (CPD 5.87 sticks fewer) compared with male smokers and (iii) cigarette affordability was significantly lower (RIP 4.32 pp higher) among smokers with lower level of education (primary school and less, RIP 4.32 pp higher; middle and high school, RIP 1.83 higher) compared with smokers with higher level of education (college and above). However, no statistical significant difference in daily cigarette consumption was found across education levels; (iv) with respect to geographical locations, affordability was significantly lower among those living in the Northeast (RIP 0.78 pp higher), Central (RIP 2.20 pp higher) and Western (RIP 3.03 pp higher) regions compared with those living in the Eastern region. Not surprisingly, smokers in those regions consumed fewer cigarettes (CPD 2.53 sticks fewer for the Northeast smokers; CPD 0.80 sticks fewer for the Central smokers without statistical significance and CPD 3.10 sticks fewer for the Western smokers) compared with those living in the Eastern region; (v) cigarette affordability (RIP 3.48 pp lower) was significantly higher among urban smokers compared with rural smokers, however, no statistical difference was found in daily consumption; (vi) luxury and premium brand cigarettes were less affordable (RIP 4.63 pp higher for Tier I, 1.05 pp higher for Tier II without statistical significance) and discount brand cigarettes were more affordable compared with the medium-priced brands (RIP 1.89 pp lower for Tier IV, 4.89 pp lower for Tier V than Tier III). Additionally, smokers who purchased luxury or premium brands tend to consume fewer cigarettes per day (CPD 2.91 sticks fewer for Tier I smokers, 3.67 sticks fewer for Tier II smokers), and smokers who use discount brands tend to smoke more compared to those using medium-priced brands (CPD 1.11 sticks more for Tier IV smokers and 2.55 sticks more for Tier V smokers compared with Tier III smokers).

### 3.3. The Association between Cigarette Affordability and Cigarette Consumption

Table 3 presents the estimated correlation between cigarette affordability and cigarette consumption. Models 1–5 adopted the per capita household disposable income as income variable to define RIP, and Model 6 adopted the personal income as income variable to define RIP. Model 1, shown in the first column of Table 3, considered only the key explanatory variable, cigarette affordability, Model 2, Model 3, Model 4 and Model 5 added, sequentially, control variables including demographics and socioeconomic status, geolocations, cigarette brands by price tier and survey wave. The results show that there were few changes in the regression coefficients and standard error across Model 2, Model 3, Model 4 and Model 5. The key findings in Table 3 include: (i) the overall conditional cigarette affordability elasticity of demand clustered around −0.17, implying that a 10% increase in the RIP (indicates 10% decrease of cigarette affordability) would result in a reduction in CPD among middle-aged and elderly Chinese smokers by 1.7%. (ii) Further investigation of subgroups show that older age and females had a lower level of cigarette consumption comparing with younger and male counterparts. (iii) Lower level of education was associated with higher level of cigarette consumption. (iv) Smokers resided in the the Northeast region, Western region and urban area were found to be associated with lower level of cigarette consumption. (v) Luxury brand cigarettes (Tier I) were associated with lower consumption, while the discount brands (Tier IV and V) were associated with a higher level of consumption. (vi) The results in model 6 were very close to the main estimates in model 5, implying that our results were robust with regard to different measures of RIP. 

Using Model 5 in Table 3 as the baseline model, we furtherly examined how cigarette consumption may respond to affordability changes differentially by age, gender, education level, marital status, geolocations and cigarettes price tier consumed (Table 4). The estimated coefficients for the interaction terms were statistically significant except for Model 5b and Model 5d, indicating that the conditional affordability elasticities of demand differed across age, education level, economic region, residential zone and cigarette brands by price tier. In other words, while smokers in general would respond to cigarette affordability decrease by reducing the number of cigarettes smoked per day, smokers aged 60 years and above would reduce their cigarette consumption more than those aged 45–59 years. Smokers with primary education and below would reduce their cigarette consumption more than those with a high level of education (college and above). Smokers living in the Northeast region would reduce their cigarette consumption less than those in the Eastern region. Urban smokers would reduce cigarette consumption less than rural smokers, with other things being constant. In addition, smokers consumed medium-priced and discount brand cigarettes would reduce their consumption less than those that smoked luxury brand cigarettes. The consistent results estimated by using the measure #2 of RIP are presented in Table A4.

Given the significant differences in conditional cigarette affordability elasticities of demand by age and education subgroups, geolocations (economic region and residential zone), and cigarette price tiers, we estimated the elasticities separately by subgroups (Table 5).

The estimated affordability elasticity was −0.109 for smokers aged 45–59 years and −0.218 for smokers aged 60 years and above, respectively. It implies that older smokers’ cigarette consumption was more responsive to changes in cigarette affordability than that of middle-aged smokers. The estimated affordability elasticity for smokers with a college degree did not statistically differ from 0, meaning that those smokers may not significantly reduce their daily cigarette consumption when cigarette affordability decreases. The estimated conditional cigarette affordability elasticities of demand based on lower education subsamples ranged from −0.129 to −0.194, with the 95% CIs overlapping.

The magnitude of the estimated conditional cigarette affordability elasticities of demand by economic regions ranged from −0.084 to −0.183, with 95% CIs overlapping across the four regions. The estimated affordability elasticity for smokers living in the Northeast region was the lowest, meaning that those smokers reduce less daily cigarette consumption than smokers in other regions when cigarette affordability decreases. Moreover, the estimated conditional cigarette affordability elasticities of demand were −0.130 for urban smokers and −0.191 for rural smokers, with overlapping 95% CIs.

Smokers who consumed the luxury brands (Tier I) had the highest estimated affordability elasticity, meaning that those smokers would reduce daily cigarette consumption most when cigarette affordability decreases. However, the estimated affordability elasticity for smokers who consumed the premium brands (Tier II) did not statistically differ from 0, meaning that those smokers may not significantly reduce their daily cigarette consumption when cigarette affordability decreases. The estimated affordability elasticities of demand for smokers who consumed medium-priced and discount brands (Tier III, Tier IV and Tier V) ranged from –0.118 to –0.192, with the 95% CIs overlapping.

Additionaly, the estimates by gender and marital status are shown in Appendix A (Table A3). The estimated affordability elasticities for male and female smokers were –0.172 and –0.095, respectively. For unmarried and married smokers, the estimated affordability elasticity was –0.159 and –0.224, respectively. The cigarette affordability elasticities of demand estimated by the measure #2 of RIP for all subsamples are shown in the Appendix B (Table A5). Those results were basically consistent with the estimates predicted by the measure #1 of RIP.

## 4. Discussion

Our analysis of the CHARLS data found that cigarette affordability (reversal of RIP) was positively associated with cigarette consumption among middle-aged and elderly Chinese smokers. Specifically, the estimated overall conditional cigarette affordability elasticity of demand was −0.165, implying a 10% decrease of cigarette affordability would result in a reduction in cigarette consumption by 1.65%. Additionally, using personal income as an alternative measure of RIP resulted in a similar elasticity value, which was −0.230, implying that both analytic approaches yield consistent results. This result is in accordance with Huang et al. [25], which found that the conditional cigarette demand price elasticity ranged from −0.12 to −0.14 among Chinese adult smokers. In addition, it is notable that the correlations between cigarette affordability and consumption are varied when examining various categories including age and education levels, geolocations (economic region and residential zone), and cigarette price tiers.

### 4.1. Association between Affordability and Consumption, by Demographics and Socialeconomic Status

We found that cigarette affordability was higher among those with high level of education. This finding is consistent with the results of Nargis et al. [26], which showed that smokers in higher socioeconomic status (SES) demonstrated greater affordability of cigarettes than those with lower SES. Furthermore, although cigarettes were more affordable for smokers with a high level of education, they tended to smoke fewer cigarettes. This finding may be explained by better knowledge and perception of health risks associated with smoking among smokers with a high level of education, which in turn could boost their self-efficacy to resist the urge to smoke [27].

Specifically, smokers aged 60 years and above was significantly more sensitive to changes in cigarette affordability than that of smokers aged 45–59. In other words, when cigarette affordability decreases, all smokers would reduce their cigarette consumption, however, smokers aged 60 years and above would reduce their cigarette consumption more than smokers aged 45–59 years. In addition, smokers with low level of education (less than high school) were significantly more sensitive to changes in cigarette affordability than that among smokers with bachelor’s degree and above. It indicates that when cigarette becomes less affordable, all smokers reduce their consumption, however, smokers with low level of education would reduce their cigarette consumption more than smokers with high level of education.

One of the possible explanations for the above results may be the exclusion of the wealth effect in our analysis. Wealth can influence consumer behavior. Chinese smokers with low level of education and elderly Chinese smokers were more sensitive to changes in cigarette affordability may be because they have less accumulation of wealth due to limited financial means, inadequate perceptions and knowledge on investments, etc., which result in a more sensitive response to changes in cigarette affordability compared with well-educated and younger smokers. Since both wealth accumulation and income flow influence affordability, future studies, particularly policy simulation analysis, would benefit from incorporating both disposable income and accumulated wealth into measuring consumers’ purchasing power.

### 4.2. Association between Affordability and Consumption, by Geographic Locations

We also found that, compared with smokers living in the Eastern region, smokers living in the Northeast region were significantly less sensitive to changes in cigarette affordability. That is, smokers living in the Northeast region may not significantly reduce their daily cigarette consumption when cigarette affordability decreases. The Northeast region was the earliest industrialized area in China, the three Northeastern rust belt provinces—Liaoning, Jilin and Heilongjiang—have been recorded among the slowest rates of growth for decades. Economic recession increases the amount and intensity of stress that people have to deal with [28], and stress decreases the ability to resist smoking and potentiates smoking intensity [29]. Thus, it is possible that economic crisis, causing psychosocial stress, diminished the effect of the cigarette affordability decrease on consumption. This suggests that these middle-aged and elderly smokers living in the Northeast China have been benefited less from the national tobacco control policies, such as the tobacco tax/price increases. To address this issue, comprehensive smoke-free laws could be considered. In addition to reducing air pollutants and protecting non-smokers from exposure to cigarette smoke, smoke-free laws have been positively associated with quitting among smokers [30]. This is especially important in China, where smoking has long been associated with a positive image and is embedded in many social events and customs [31]. However, China does not have a national smoke-free law yet. Furthermore, among the thirteen cities achieved 100% smoke-free by smoke-free regulations (or laws), there are only two cites (Changchun and Anshan) in Northeast region.

In addition, this study found that cigarettes were less affordable for rural smokers than for urban smokers, however, the former tended to smoke more cigarettes per day than the latter. This result may be explained by the knowledge gap between urban and rural smokers, with the former being more aware of the health consequences of smoking than the latter. According to the GATS (2015), the percentage of adult smokers who believe that smoking causes stroke, myocardial infarction, lung cancer or erectile dysfunction were 7.6% in rural areas and 14.4% in urban areas in China [10]. Moreover, in fourteen countries where GATS was conducted during 2008–2010, the awareness about smoking tobacco being a cause of serious illness was the least in Chinese elderly adults (65.6%) and rural residents (74.5%) across all selected countries [32]. As such, tobacco control advocates should be strengthened in rural China. Specifically, rural smokers were significantly more sensitive to changes in cigarette affordability than urban smokers. In other words, when cigarette affordability decreases, rural smokers would reduce more cigarette consumption than urban smokers. This suggests that tax/price measures could help to reduce the disparity in tobacco consumption and improve the health equity between rural and urban smokers.

### 4.3. Association between Affordability and Consumption, by Cigarette Price Tiers

Our study reveals that smokers who used luxury or premium brands tend to smoke less and smokers who used discount brands tend to smoke more per day compared with those who used medium-priced brands. This is consistent with Xu et al. (2019), they found that those who used premium cigarette brands were more likely to be non-daily smokers, and those who used medium-priced brands were less likely to be daily smokers compared with those who used discount brands [33]. With respect to cigarette affordability, cigarettes were more affordable for those discount brand smokers and less affordable for those luxury or premium brand smokers compared with medium-priced brand smokers, which is consistent with the finding observed by Nargis et al. in their study on cigarette affordability in China [26]. Additionally, we found that those deep discount brand (Tier V) cigarette smokers were less responsive to cigarette affordability change compared to smokers consumed cigarettes of higher price tiers. These findings imply that the extremely low price of discount cigarette brands in China enabled their targeted consumers to enjoy high affordability of cigarettes, and hence less likely to reduce their cigarette consumption when cigarette affordability decreased. To address this issue, the Chinese government can establish a minimum price for cigarettes to ensure the discount cigarette brands become less affordable. The minimum pricing policy could be beneficial for protecting vulnerable smokers, particularly youth and low-income smokers.

One major limitation of this study was that we only considered the impact of cigarette affordability on cigarette consumption among those who continued to smoke. It left out the impact of decrease in cigarette affordability on promoting quitting among smokers and on discouraging smoking initiation among non-smokers. It has been demonstrated that the future benefits of non-smoking are substantial, particularly for low-income groups, and they outweigh any potential negative impact that may result from higher tobacco taxes, which reduce cigarette affordability [34,35]. If the smoking cessation benefits and the long-term benefits on reducing smoking initiation were included, the beneficial impact of reducing cigarette affordability would be much larger than the impact estimated in this study, which only considered the effect on reducing cigarette consumption among older Chinese adults.

## 5. Conclusions

In October 2016, President Xi Jinping announced the Healthy China 2030 blueprint setting public health as a precondition for all future economic and social development. Three years later in July 2019, a corresponding action plan was released by the Chinese State Council. The plan encompassed 15 goals to be achieved between 2020–2030 with specific targets, which include decreasing the health effect of smoking, preventing chronic diseases, implementing the elderly health promotion initiative, etc. Given the rapidly rising incidence of non-communicable diseases [36,37], as well as a rapidly ageing population in China [1], the release of the plan is both timely and important. Of particular importance, the plan focuses on the promotion of public health and disease prevention, which represents a strategic shift from treatment to prevention [38]. Several major findings in our study could provide timely evidence to policy makers to support the tobacco control measures outlined in the Healthy China 2030 action plan: (i) Healthy China 2030 action plan recognizes tobacco taxation and pricing as an important tobacco control measure to reduce tobacco use. This study provides strong evidence that tax/pricing policies that reducing cigarette affordability can help curb cigarette consumption among adults, as well as elderly smokers who are more likely to be affected by NCDs compared to their younger counterparts [17]. Importantly, raising cigarette price and thus making cigarettes less affordable could have a larger impact on reducing the number of cigarettes smoked for elderly smokers, smokers with low-level of education and rural smokers, which could in turn benefit these vulnerable groups by reducing their health burden and economic costs attributed to smoking. (ii) Additionally, we found that smokers living in the Northeast region were significantly less sensitive to changes in cigarette affordability, which suggests other tobacco control measures, such as smoke-free legislation, could be used in the Northeast provinces to complement and strengthen the impact of tobacco tax/pricing policies. (iii) Given the lack of sensitivity to changes in affordability among smokers who use discount brands, it may be important to adopt minimum price regulations and restrictions on price promotions to reduce price variability across different brands. This is critically important for the benefit of protecting vulnerable smokers, such as youth and low-income smokers. 

## Figures and Tables

**Table 1 ijerph-16-04832-t001:** Descriptive statistics of the analytical sample.

Sample Information	Wave 1 (2011–2012)		Wave 3 (2015–2016)		Pooled	
	N	%	N	%	N	%
Overall	3018	100%	3634	100%	6652	100%
Age group						
45–59	1687	56%	1817	50%	3504	53%
60+	1331	44%	1817	50%	3148	47%
Gender						
Male	2760	91%	3322	91%	6082	91%
Female	258	9%	312	9%	570	9%
Education level						
Primary and below	1685	56%	1945	54%	3630	55%
Middle and high school	1260	42%	1594	44%	2854	43%
College and above	73	2%	95	3%	168	3%
Marital status						
Unmarried	265	9%	368	10%	633	10%
Married	2753	91%	3266	90%	6019	90%
Economic Region						
Eastern region	959	32%	1157	32%	2116	32%
Northeast region	220	7%	283	8%	503	8%
Central region	963	32%	1128	31%	2091	31%
Western region	876	29%	1066	29%	1942	29%
Residential zone						
Rural	1760	58%	2101	58%	3861	58%
Urban	1258	42%	1533	42%	2791	42%
Cigarette brands by price tier						
Tier I (luxury brands)	56	2%	164	5%	220	3%
Tier II (premium brands)	42	1%	156	4%	198	3%
Tier III (medium-priced brands)	531	18%	1481	41%	2012	30%
Tier IV (discount brands)	1577	52%	1599	44%	3176	48%
Tier V (deep discount brands)	812	27%	234	6%	1046	16%

**Table 2 ijerph-16-04832-t002:** Generalized estimating Equation (GEE) estimates of relative income price and cigarettes consumed per day.

Characteristic	RIP	Cigarettes Consumed Per Day
Age ≥ 60 (vs. Age 45–59)	1.603 ***	−2.407 ***
	(0.301)	(0.391)
Female (vs. male)	0.820	−5.873 ***
	(0.504)	(0.527)
Primary and below (vs. college and above)	4.320 ***	1.020
	(0.776)	(1.186)
Middle and high school (vs. college and above)	1.827 **	1.156
	(0.727)	(1.146)
Married (vs. unmarried)	−1.098 *	0.318
	(0.637)	(0.538)
Northeast region (vs. Eastern region)	0.776 *	−2.533 ***
	(0.440)	(0.743)
Central region (vs. Eastern region)	2.198 ***	−0.801
	(0.353)	(0.558)
Western region (vs. Eastern region)	3.029 ***	−3.103 ***
	(0.408)	(0.546)
Urban (vs. rural)	−3.484 ***	−0.262
	(0.317)	(0.418)
Tier I (vs. tier III)	4.633 ***	−2.905 ***
	(0.875)	(0.957)
Tier II (vs. tier III)	1.047	−3.665 **
	(0.966)	(1.560)
Tier IV (vs. tier III)	−1.893 ***	1.110 **
	(0.360)	(0.468)
Tier V (vs. tier III)	−4.890 ***	2.546 ***
	(0.487)	(0.664)
Wave 3 (vs. wave 1)	0.610**	0.520
	(0.278)	(0.381)
Constant	6.794 ***	18.080 ***
	(1.013)	(1.393)
Number of observations	*N* = 6652	
Number of groups	*N* = 5077	

Note: (1) The statistics reported are the coefficients of independent variables with robust standard errors in parenthesis. (2) ***, ** and * indicate statistical significance at the 1%, 5% and 10% level, respectively.

**Table 3 ijerph-16-04832-t003:** GEE estimates of the association between cigarette affordability and the average number of cigarettes consumed per day.

Variable	Measure #1					Measure #2
	Model 1 Coef. (S.E.)	Model 2 Coef. (S.E.)	Model 3 Coef. (S.E.)	Model 4 Coef. (S.E.)	Model 5 Coef. (S.E.)	Model 6 Coef. (S.E.)
Number of observations	*N* = 6652					*N* = 4257
Number of groups	*N* = 5077					*N* = 3505
Ln (RIP)	−0.178 ***	−0.180 ***	−0.184 ***	−0.165 ***	−0.165 ***	−0.230 ***
	(0.016)	(0.016)	(0.016)	(0.015)	(0.016)	(0.016)
Age ≥60 (age 45–59 = 0)		−0.138 ***	−0.142 ***	−0.157 ***	−0.164 ***	−0.205 ***
		(0.031)	(0.031)	(0.029)	(0.029)	(0.038)
Female (male = 0)		−0.479 ***	−0.463 ***	−0.483 ***	−0.487 ***	−0.384 ***
		(0.051)	(0.052)	(0.052)	(0.052)	(0.076)
Primary and below (college and above = 0)		0.294 ***	0.244 ***	0.177 *	0.172 *	0.225 **
		(0.090)	(0.091)	(0.096)	(0.096)	(0.091)
Middle and high school (college and above = 0)		0.244 ***	0.207 **	0.154 *	0.150	0.226 ***
		(0.090)	(0.090)	(0.093)	(0.094)	(0.086)
Married (unmarried = 0)		−0.003	0.001	0.005	0.005	−0.030
		(0.045)	(0.044)	(0.044)	(0.045)	(0.060)
Northeast region (Eastern region = 0)			−0.096 *	−0.114 **	−0.117 **	−0.153 **
			(0.055)	(0.054)	(0.053)	(0.062)
Central region (Eastern region = 0)			0.009	−0.014	−0.018	−0.063
			(0.039)	(0.037)	(0.037)	(0.043)
Western region (Eastern region = 0)			−0.133 ***	−0.152 ***	−0.156 ***	−0.104 **
			(0.039)	(0.037)	(0.036)	(0.043)
Urban (rural = 0)			−0.094 ***	−0.064 **	−0.059 **	−0.102 ***
			(0.031)	(0.029)	(0.029)	(0.035)
Tier I (Tier III = 0)				−0.130	−0.133 *	−0.085
				(0.079)	(0.079)	(0.091)
Tier II (Tier III = 0)				−0.160	−0.168	−0.086
				(0.106)	(0.108)	(0.133)
Tier IV (Tier III = 0)				0.048	0.064 *	0.072 **
				(0.032)	(0.034)	(0.036)
Tier V (Tier III = 0)				0.101 ***	0.134 ***	0.222 ***
				(0.038)	(0.045)	(0.048)
Wave 3 (wave 1 = 0)					0.060 **	0.043
					(0.027)	(0.034)
Constant	2.086 ***	1.923 ***	2.042 ***	2.133 ***	2.097 ***	1.846 ***
	(0.047)	(0.109)	(0.113)	(0.115)	(0.116)	(0.132)
Wald $\chi^{2}$	124.91	252.27	332.89	374.99	395.56	510.55

Note: (1) The statistics reported are the coefficients of independent variables with robust standard errors in parenthesis. (2) ***, ** and * indicate statistical significance at the 1%, 5% and 10% level, respectively. (3) For measure #1 (from model 1 to model 5), relative income price (RIP) was measured according to the per capita household disposable income; for measure #2 (in model 6), RIP was measured according to the personal income. (4) Results were estimated by Equation (3).

**Table 4 ijerph-16-04832-t004:** Differential responsiveness of cigarettes per day (CPD) to cigarette affordability.

Variable	Model 5a Coef. (S.E.)	Model 5b Coef. (S.E.)	Model 5c Coef. (S.E.)	Model 5d Coef. (S.E.)	Model 5e Coef. (S.E.)	Model 5f Coef. (S.E.)	Model 5g Coef. (S.E.)
ln(RIP)	−0.099 ***	−0.170 ***	0.069	−0.207 ***	−0.163 ***	−0.193 ***	−0.185 ***
	(0.023)	(0.017)	(0.102)	(0.043)	(0.035)	(0.017)	(0.034)
**Interaction terms**							
ln(RIP) × Age≥60	−0.126 ***						
	(0.029)						
ln(RIP) × Female		0.046					
		(0.043)					
ln(RIP) × Primary and below			−0.269 ***				
			(0.104)				
ln(RIP) × Middle and high school			−0.193 *				
			(0.105)				
ln(RIP) × Married				0.048			
				(0.046)			
ln(RIP)× Northeast region					0.093 *		
					(0.056)		
ln(RIP)× Central region					−0.012		
					(0.042)		
ln(RIP)× Western region					−0.017		
					(0.043)		
ln(RIP)× Urban						0.064 **	
						(0.032)	
ln(RIP)× Tier I							−0.291 ***
							(0.109)
ln(RIP)× Tier II							0.022
							(0.196)
ln(*RIP*) × *Tier* IV							0.023
							(0.040)
ln(*RIP*) × *Tier* V							0.074 *
							(0.042)
Other variables	Control	Control	Control	Control	Control	Control	Control
Wald $\chi^{2}$	439.94	396.19	410.22	395.51	412.75	442.23	413.23
Number of observations	*N* = 6652						
Number of groups	*N* = 5077						

Note: (1) The statistics reported are the coefficients of independent variables with robust standard errors in parenthesis. (2) ***, ** and * indicate statistical significance at the 1%, 5% and 10% level, respectively. (3) The results for other independent variables in regression are not included in the table due to word limit. Such variables include age group (Age ≥ 60), gender (female), education level (primary and below, middle and high school), marital status (married), economic regions (Northeast region, Central region and Western region), residential zone (urban), cigarette price tiers (Tier I, II, IV, V) and survey wave (wave 3). (4) Results were estimated by Equation (4).

**Table 5 ijerph-16-04832-t005:** Conditional cigarette affordability elasticities of demand by subgroups.

Characteristic	Observations	ln(RIP)	95% CIs	Other Variables
Age group				
45–59	3504	−0.109 ***	(−0.154 to −0.064)	Control
60 +	3148	−0.218 ***	(−0.258 to −0.178)	Control
Education level				
Primary and below	3630	−0.194 ***	(−0.232 to −0.156)	Control
Middle and high school	2854	−0.129 ***	(−0.179 to −0.079)	Control
College and above	168	0.048	(−0.205 to- 0.301)	Control
Economic Region				
Eastern region	2116	−0.171 ***	(−0.234 to −0.107)	Control
Northeast region	503	−0.084 *	(−0.174 to 0.005)	Control
Central region	2091	−0.167 ***	(−0.213 to −0.121)	Control
Western region	1942	−0.183 ***	(−0.233 to −0.133)	Control
Residential zone				
Rural	3861	−0.191 ***	(−0.224 to −0.158)	Control
Urban	2791	−0.130 ***	(−0.186 to −0.075)	Control
Cigarette brands by price tier				
Tier I	220	−0.499 ***	(−0.728 to −0.270)	Control
Tier II	198	−0.136	(−0.380 to 0.107)	Control
Tier III	2012	−0.192 ***	(−0.255 to −0.130)	Control
Tier IV	3176	−0.153 ***	(−0.193 to −0.113)	Control
Tier V	1046	−0.118 ***	(−0.169 to −0.068)	Control

Note: (1) The statistics reported are the coefficients of independent variables. (2) *** and * indicate statistical significance at the 1%, 5% and 10% level, respectively. (3) The results for other independent variables in regression are not included in the table due to word limit. Such variables include age group (Age ≥ 60), gender (female), education level (primary and below; middle and high school), marital status (married), economic regions (Northeast region, Central region, Western region), residential zone (urban), cigarette price tiers (Tier I, II, IV, V) and survey wave (wave 3). (4) The control variables are same within one subgroup, but different among subgroups. Control variables used for grouping were not included in each of the subgroup regressions. For example, we exclude the age group (Age ≥ 60) variable when performing estimates by age subgroups.

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
