# Peer review of "Cigarette Affordability and Cigarette Consumption among Adult and Elderly Chinese Smokers: Evidence from A Longitudinal Study"

_ijerph, 2019, doi:10.3390/ijerph16234832_

Round 1

Reviewer 1 Report

This is a well written article on an issue of global public health concern, but with particular relevance to developing nations that find themselves facing a similar problem with tobacco that Western nations faced 50 years ago. I have several points of guidance for the authors.

Smoking in China:

With its large population and relatively high rate of smoking, especially among men, China stands to inherit a large health burden from tobacco related illness. Public health measures that educate the population about the dangers of smoking, limit smoking in public spaces, curb smoking uptake among young people, and encourage smoking reduction and cessation among current smokers, will go a long way towards reducing the burden of smoking on China’s health system.  Taxation is only one part of the public health solution and needs to be understood as such.

Your introduction should mention that smoking in China is significantly male dominated. Providing overall smoking rates without a gender-split is misleading, as an uninitiated international reader might assume a relatively even distribution. This is especially important when you later show 91% of over-45 smokers in the CHARLS study are male.

Price as public health policy:

Taxation to raise cigarette retail prices is one of the simplest tools in the public health armoury to help discourage smoking uptake and encourage cessation among current smokers. Taxation-based strategies can deliver substantial public funds to government, depending on the rate of taxation, and these funds can be used to support smokers in the health system and to support other public health activities around smoking cessation. Unfortunately, many governments also place these funds into consolidated revenue and use them for non-health related purposes.

Cigarette price as a public health strategy has several flaws. Price increases tend to work best among low-income populations for reasons of relative affordability. This strategy discriminates against poorer smokers and among those suffering from common mental illnesses such as stress, anxiety, and depression who find it much harder to give up smoking due to their illness.

Assumptions about smokers:

There is a changing profile of smokers in advanced nations with existing public health campaigns around smoking cessation.  Such nations have seen reductions in smoking uptake and increases in quitting among their general populations over the past 30-plus years. However, over the same period they have observed elevated uptake and persistence of smoking among the sector of population experiencing common mental health disorders such as stress, anxiety, and depression. This has occurred despite vastly increased tobacco taxation causing large price rises in cigarettes. As such, you may want to mention in that you cannot be sure how cigarette price rises will change the smoking rates among Chinese with ongoing mental illness.

See: Hirono KT and Smith KE. (2018) Australia’s $40 per pack cigarette tax plans: the need to consider equity. Tobacco Control 27: 229.

Several times you make the assertion that older smokers are more addicted to nicotine because they have smoked for longer, and therefore it will be harder to influence their smoking behaviours. This is simply untrue. A smoker’s level of addiction to nicotine is unrelated to their length of habit, once that habit has been in place more than a few years. A 25-year old smoker of 10 years can be just as addicted to nicotine as a 65-year old smoker of 50 years. The main reason why younger AND older smokers are still smoking is because they have not been incentivized enough to reduce or give-up smoking. Tobacco excise, education, place-based restrictions, plain packs with graphic warnings, etcetera…. are required to act in unison to reduce smoking uptake and continuation of smoking, regardless of age or length of habit.

Measures:

I disagree with the use of per capita household disposable income as the foundation for understanding affordability of cigarettes for individual smokers. This is misleading, and almost certainly distorts the true relationship between cigarette pricing and willingness to pay. It assumes all income can be shared equally among those living in the same household and can be used preferentially to purchase cigarettes for the member/s of the household who are smokers at the possible expense of purchasing decisions of non-smoking financial contributors to the household. Given that over 90% of your smokers are male, the chosen income measure disregards the purchasing decisions and autonomy of women in the household who may also be contributing financially, whatever their age.

Cigarette affordability is a key concept underlying this paper. To have greater credibility in their affordability measure the authors should use only the income of those identified as smokers in the household, and only at the level of the individual. Otherwise, these results may not be transferable to real world outcomes should Chinese public health officials use this paper as a basis for their cigarette taxation policy.

The focus on those aged 45-plus is driven by the CHARLS data source beginning at age 45 years, but the authors could do more to acknowledge that smoking uptake mostly occurs in the teenage years. A 45-year-old smoker may have already smoked for 30 years. For those aged 60-plus who have smoked for most of their lives, it is likely that many already have the beginnings of smoking related illness, even if they reduce CPD or give up smoking altogether. So, while it is interesting to consider how taxation affects CPD in these older people, there is likely more to be gained from a public health cost reduction perspective from concentrating on those in the 45-59 years age group (and younger, but that is outside the scope of CHARLS).

There are some minor English language issues throughout this paper that need to be addressed. Please have an English language editor assess your paper for grammar and spelling.

Reviewer 2 Report

The paper suggest using the CHARLS data results in an estimation of affordability. This study claim to examine the potential differential impact of cigarette affordability on cigarette consumption by population subgroups, defined by demographics and socioeconomic status. However, the article fails to capture variation in brands, so variations in quality. Failing to keep constant prices as in Blecher et al 2004, or He et al 2018, or even in the authors previous publication Cigarette Affordability in China: 2001-2016, results in a unclear combination of affordability + something else. This means this is not an analysis of affordability but an analysis of prices paid by different smokers along the time.

This being said, the article combine price data and show association between affordability and consumption. Due the limitation of the data and the analysis, interpretations on the potential impact of reduction of affordability on consumption should be restricted to the strong assumptions behind the estimation (not descried in the manuscript). The suggestion of including wealth analysis in future research seems to be a good opportunity for further research in China and other countries.

Limitations on the impact of affordability on cessation is addressed.

The affordability of cigarette across regions and regional information on cigarette sales, appear late in the document. The same happened with other good comparisons with, for example GATS. It would be a good idea to include a description section dealing with this and other topics discussed in the document (including weights to understand how this results in relationship with the total retired population)  

Additionally, the documents seems to be in a very early stage of the process. Errors and unclear sections require a critical review before re-submission

Line 396 to 399 Please check references 8 and 9

Line 80 to 101 It is not clear why you exclude the second wave and not the first wave, that will have the same definition (or it is different, that is not clear)

Line 105. They only report 20 cigarette packs? Or 20 packs equivalent?

Line 122, please include a reference on the use of this technique to deal with outliers

Line 124, please include a reference on the use of this technique to deal with outliers

Section 2.4 is unclear. The reader has to go down to line 138 to understand what equation (1) means

What is yi?

Why multiplying for one covariate in xi in equation (4)?

Line 149 are you talking about the Independence Model Criterion (QIC)?

Line 192 Authors describe model in line 192, after finishing the discussion of methods and introducing results. This results in a confusing document

Line 204 The authors call Model 8_1 something not defined, this results in a even more confusing manuscript

Table 4 show different “Model”s with no clear identification

Round 2

Reviewer 2 Report

The authors provide a long detailed explanation to the reviewer's concerns. Authors introduced multiple changes in the manuscript resulting in an easier to read version. 

Comment #1 gives more clarity on the topic, however, it is not explaining how these tiers can solve the problem that the authors seem to observe in the data, considering its own response. Observing the data it looks like introducing tiers in the analysis has a big impact on the estimations. However, the introduction of tiers was not clearly explained. How the tiers are used considering inflation in China? does tiers change along the time? these questions have no answer in the document.

Introducing tiers as covariances seem to create een more problems. How the authors explain running a regression of RIP against tiers?

Previous comments were related with comment #2. The response suggested by the authors is completely inaccurate for the comment and is not responding to how RIP used for distribution of brands with an unknown distribution and other characteristics not captured by covariances. Responses releted on WHO using RIP for different countries are completely out of this discussion. If authors want to use this method first they need to work more on the characteristics of this price a the implications of using this to compare surveys. This is not observed in the manuscript.

Authors do not address how data and analysis limitation results in a limitation on the potential impact of the reduction of affordability on consumption. Authors must consider that results can only be restricted to the strong assumptions behind the estimation (still not described in the manuscript).
